# Construction and Usefulness Verification of Modeling Method of Subsurface Soil Layers for Numerical Analysis of Urban Area Ground Motion

**Hiroki Motoyama [1],* and Muneo Hori [2]**

1   Department of Civil and Environmental Engineering, National Defense Academy, Yokosuka 239-8686, Japan
2   Research Institute for Value-Added-Information Generation, Japan Agency for Marine-Earth Science and Technology, Yokosuka 237-0061, Japan; horimune@jamstec.go.jp
*   Correspondence: hiromoto@nda.ac.jp; Tel.: +81-46-841-3810

**Abstract:** Estimation of urban seismic damage using numerical simulation needs an automatic modeling method of surface layers and residential buildings. This study focuses on modeling of surface layers and shows a method of constructing models by interpolating boring data. An important property of the modeling method is robustness, that means that the method works for boring data with inconsistent soil layers. To satisfy this, we developed the method using artificial layers. We applied the method to a test site and checked its robustness. This test also showed that the method gave realistic models. Finally, we applied the method to the estimation of urban seismic damage and discussed the usefulness by comparing the result with one obtained by a conventional method.

**Keywords:** urban area ground motion; robust modeling method; construction of ground model; artificial layers; integrated earthquake simulation





## 1. Introduction

In Japan, it is standard practice to estimate earthquake disasters by summing up potential damages of all residential buildings in a given area for a presumed earthquake. Regression models and fragility curves are used to evaluate a certain seismic index (such as the maximum acceleration or velocity) at a building site, taking into account the attributes of the building (structure type, age, etc.), and to evaluate the possibility of structural failure using these indices [1–4]. These models and curves are easy to apply to a large number of residential buildings, because they require only a small amount of data on the configuration and mechanical properties of the surface layers and residential buildings. However, large estimation errors are inevitable since they ignore physical processes of the ground motion amplification and the structural seismic responses.

In recent years, regional earthquake simulation, that applies physics-based simulations to the target region, has been studied in order to provide more scientifically reliable estimates of earthquake hazard and disaster. The physics-based simulations (that solve the solid wave equation) numerically analyze earthquake wave propagation processes and seismic response analysis processes, respectively, for earthquake hazard and disaster estimates. Programs enhanced with high performance computing (or parallel computing) capability are developed for the regional earthquake simulation.

Ground motion components of several Hz are amplified in surface layers of several tens of meters thick (in Japan, above an engineering bedrock, that is defined as a layer of shear velocity greater than 400 m/s). This frequency range coincides with that of the first natural period of an ordinary residential building. Therefore, it is essential to accurately analyze the degree of the ground motion amplification for structural seismic response. A suitable analysis model is needed for surface layers of each residential building, that are referred to as the site surface layers hereafter.

Integrated Earthquake Simulation (IES), the development of that the authors are involved in, is a pioneering program for the regional earthquake simulation [5–7]. In IES, an analysis model of site surface layers is needed for the ground motion amplification processes, in addition to the analysis model of the crust for the earthquake wave propagation processes. Here, the analysis model refers to an input file(s) of a specific numerical analysis program that is made by converting the ground model (or a geotechnical model). It is not easy to construct a ground model for the site surface layers using a set of boring data that consist of $N$-values, types and thicknesses of each layer. Converting the ground model to the analysis model is difficult because the complex geometry and mechanical properties of each layer must be determined.

In this paper, we are aimed at proposing a methodology of robustness and automatic construction of an analysis model for the site surface layers for each residential building. It is undoubtedly better to construct a ground model of site surface layers covering $100 \times 100$ m or $1000 \times 1000$ m and to convert it to an analysis model. However, due to the limitation of the boring data available, we choose to construct an analysis model of the site surface layer to provide amplified ground motions to the structural seismic response analysis.

The contents of this paper are organized as follows. First, in Section 2, we present an overview of literature survey on current methods of constructing an analysis model for the site surface layers. Next, we describe our method of constructing an analysis model for the site surface layers; a three-dimensional artificial ground model is constructed for a small area by interpolating boring data and an analysis model for the site surface layers is constructed by determining layer properties.

## 2. Literature Survey

As explained in the preceding section, we distinguish an analysis model from a ground model. There are numerous methods for constructing a ground model for site surface layers, using observed data of natural earthquakes/seismic tremors, artificial earthquakes/vibrations or microtremors [8–10]. They range from simple models consisting of stratified layers to complex models consisting of layers of varying thickness.

The simple ground model, which is based on boring data of several tens of meters in depth, is primally used to understand ground conditions at each site. The complex ground model is often two-dimensional, to show the subsurface structure in vertical cross section. It is used to design large-scale structures, such as tunnels or subways; the two-dimensional ground model is sufficient since a two-dimensional model is used in designing such tunnel structures [11].

It is theoretically possible to construct a two-dimensional or three-dimensional ground model by interpolating boring data [12–14]. However, sufficient density of boring data is required for interpolation; higher density is needed for areas where ground conditions vary from place to place and disaster estimation is difficult in such areas.

While there are many three-dimensional geological models used for regional earthquake wave propagation simulation (that cover areas on the order of $100,000 \times 100,000 \times 10,000$ m), there are only a limited number of examples of constructing three-dimensional ground models. The largest three-dimensional ground model is constructed by Ichimura et al., which covers Tokyo Metropolis of $10,000 \times 10,000 \times 100$ m, and it is converted to an analysis model of finite element method, the degree-of-freedom of which is of the order of $100,000,000$, as it uses elements of 1 m size [15]. Such a three-dimensional ground model is effective in extracting the locations where stronger ground motion occur. However, it is a difficult task to construct the three-dimensional ground model from a set of boring data. A small scale ground model (that can be easily constructed) is sufficient for structural seismic response analysis of each residential building.

The major difficulty in constructing a ground model is the inconsistency of adjacent boring data, which refers to a difference in the sequence of layers. In alluvial plains, stratum complexity is inevitable due to uneven sedimentation action, and this inconsistency reflects

the complexity of the surface layers. It is known that application of simple kriging to inconsistent boring data often results in unrealistic ground models [12,13].

An alternative to kriging is the use of artificial layer [16]. An artificial layer has a thickness of zero, and by substituting appropriate artificial layers into boring data, it is possible to always produce consistent boring data. The procedures of substituting an artificial layer are described as follows. For given $N$ boring data, denoted by $B^1$, $B^2$... and $B^N$, we first compare $B^1$ and $B^2$. If they are inconsistent, the minimum number of artificial layers are substituted into both to make them consistent. We can repeat this process of adding inconsistent boring data of $B^i$ to a set of consistent boring data of $\{B^1, B^2, ..., B^{i-1}\}$ until all $B^i$'s are consistent. Figure 1 depicts a case of $N = 3$. The consistent $B^i$'s share a large number of surface layers, although each $B^i$ initially has a small number of surface layers. The surface layer number corresponds to the number of artificial layers to be substituted, and it increases as $N$ increase and the degree of inconsistency of $B^i$'s increases.

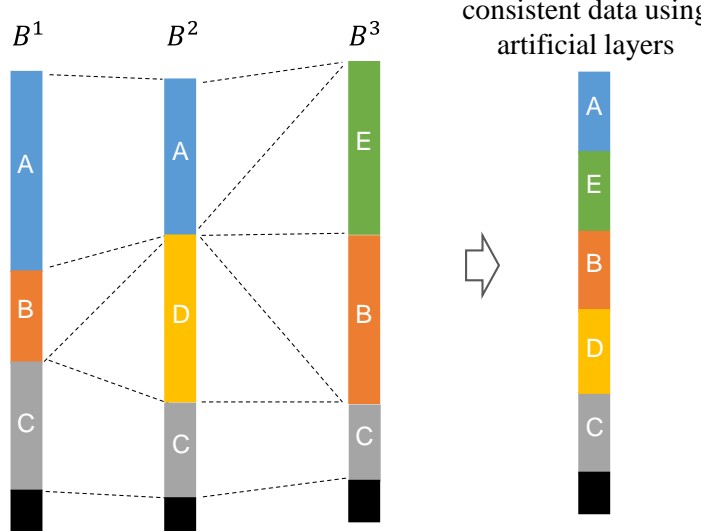

**Figure 1.** Constructing a ground model using artificial layers.

For a given ground model, we can manually convert an analysis model of finite difference method or finite element method, which solves the wave equation for the ground motion amplification processes. The conversion is performed for the configuration and mechanical properties (such as density, elasticity or wave velocity) of each surface layer. While the geometry of the ground model is readily converted, the mechanical properties of the analysis model cannot be easily determined because they are often not described in the ground model.

At the end of this section, we briefly mention the current status of IES. As for the physics-based simulation, IES has the following two core technologies; (1) numerical analysis programs of solving wave equation enhanced with high performance computing capability; and (2) programs of automatically constructing an analysis model for each numerical analysis. Currently, it is possible to automatically construct analysis models for the crust and structures for a given geological model and digital data, respectively [15,17,18]. What remains is the surface layer, for which no ground model exists. Therefore, it is necessary to develop a method to construct an analysis model of the site surface layer that links the wave propagation analysis using the crust analysis model and the seismic response analysis using the structure analysis model.

## 3. Method of Constructing Analysis Model for Site Surface Layer

The proposed method for constructing an analytical model of the site surface layer involves the following two steps: (1) automatic construction of a three-dimensional artificial ground model, and (2) automatic construction of an analysis model of the site surface layer. The construction of the three-dimensional artificial ground model is performed using a

set of inconsistent boring data. The mechanical properties of each layer are determined by using empirical relations with *N*-values.

The main feature of the proposed method is its high robustness, i.e., it can be applied to any boring data set in the region of interest. The inconsistency of the boring data can be dealt with by using the artificial layer method described in the previous section. The non-uniform distribution of boring locations is the bottleneck of the robustness of the method. In this method, instead of using all the available boring data, we try to extract a few locations close to the target residential building and construct a three-dimensional artificial ground model. However, the robustness comes at the expense of the reliability of the constructed analysis model because some analysis models are constructed using boring data far from the site.

### 3.1. Three-Dimensional Artificial Ground Model

In the proposed method, the artificial layer method is applied to deal with inconsistent boring data. This method is not complex and can be easily applied to any set of boring data. When applying the artificial layer method, it is necessary to perform data cleansing on a given set of boring data. In other words, the boring data are converted to the same format and fluctuations in terminology (especially soil names) are eliminated.

For a consistent set of boring data, a three-dimensional artificial geotechnical model can be constructed by interpolating the data. There are many methods of interpolation, such as kriging. Advanced interpolation methods require the determination of several parameters, and the results of interpolation depend on these parameters.

To increase robustness, we apply the simplest linear interpolation and always use the set of three boring data closest to the building site. As mentioned, there is possibility that the increase in the number of boring data irrationally causes the increase in the number of artificial layers, that could violate the robustness. The interpolation is formulated as follows: denoting by $P^i$ the location of the *i*-th boring point and denoting by $f^i$ the value of the *i*-th boring data, the interpolated value at point $P$ is given as

$$f(P) = \sum_{i=1}^{3} w^i(P) f^i$$

where $w^i$ is the weight; for instance, $w^i$ is the ratio of the area of triangle $AP^2 P^3$ to the area of triangle $P^1 P^2 P^3$. We interpolated the thickness of the layer, not the elevation above and below the layer, using the above equation.

It is not necessary to interpolate for the *N*-values of the boring data. If the *N*-values are slightly different for each boring data and for each depth, we calculate the average value and assign it to that layer. However, if the *N*-values vary greatly in a common layer, robust interpolation is required. Assigning *N*-values to an artificial layer of zero thickness is not easy for this case. We use a weighted average of the N values as follows

$$N(P) = \sum_{i=1}^{3} \frac{1}{\left| P - P^i \right|} N^i$$

where $N^i$ is the *N*-value of the *i*-th boring data and $\left| P - P^i \right|$ is the distance between $P$ and $P^i$. This interpolation can be applied to the case where an artificial layer of zero thickness is included. The functions of this work are the first trial functions, and they might be improved through the validation of the models of future works.

### 3.2. Analysis Model of Site Surface Layer

A stratified layer model is employed as an analysis model, since we apply one-dimensional numerical analysis of solving wave equation of horizontal shear waves for the ground motion amplification processes. The numerical analysis is linear at this moment but is readily extended to non-linear if shear strain becomes large.

The stratified layer model can be easily constructed from the three-dimensional artificial ground model. The data of elevation of the site where the target building is located are required. The composition of the analytical model can be determined from the thickness of each layer. The stratified model requires shear wave velocity as a mechanical property. We use the following empirical equation that relates $N$-value to shear wave velocity [19].

$$V_s = \begin{cases} 80N^{1/3} & \text{for sandy soil and gravel soil,} \\ 100N^{1/3} & \text{for cohesive soil.} \end{cases}$$

To apply this equation, we must classify the soil type of the boring data into two categories of sandy soil and cohesive soil.

## 4. Example of Automatic Construction of Analytical Model of Site Surface Layer

In this section, we present an example of automatic construction of an analysis model of the site surface layer used for the physics-based simulation of ground motion amplification processes. Specifically, we use actual boring data to verify the robustness of the proposed method. The reliability of the analysis model is also verified using the first natural period of the model.

Figure 2 shows the target area of this example, which is a part of Takamatsu City, Kagawa Prefecture, Japan. The blue squares indicate residential buildings, and the red circles indicate boring locations. The number of residential buildings is 104,064.

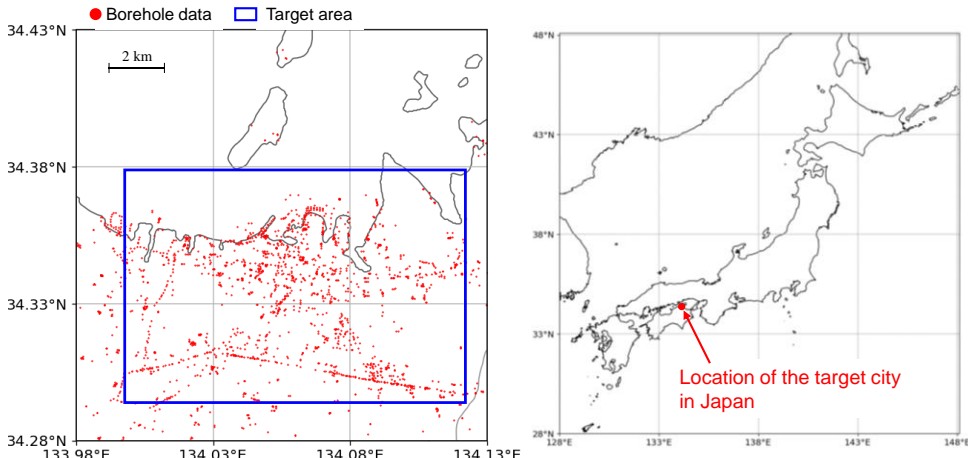

**Figure 2.** Target area (blue square area) and distribution of borehole data (red points).

First, for each residential building, the three boring data closest to the building were selected, and a three-dimensional artificial ground model was constructed. Then, an analysis model of stratified layers was constructed for the site surface layer from the artificial ground model. The following five processes were automatically performed: (1) selection of the closest boring data; (2) interpolation of layer thickness; (3) interpolation of $N$-values; (4) conversion of $N$-values to shear wave velocity: and (5) construction of the stratified layer model. It was confirmed that there were no errors in the automatic construction of the analysis model for all 104,064 residential buildings.

Figure 3 shows the distribution of the natural period of the automatically constructed analysis models. The natural period is calculated by the following equation

$$T = \sum_i \frac{4H^i}{V_s^i},$$

where $H^i$ and $V_s^i$ are the thickness and shear wave velocity of the $i$-th layer of the analysis model. The locations with long natural period appear along the coastline and rivers. This

indicates that this distribution is realistic because the ground at these locations is generally composed of thicker and softer layers.

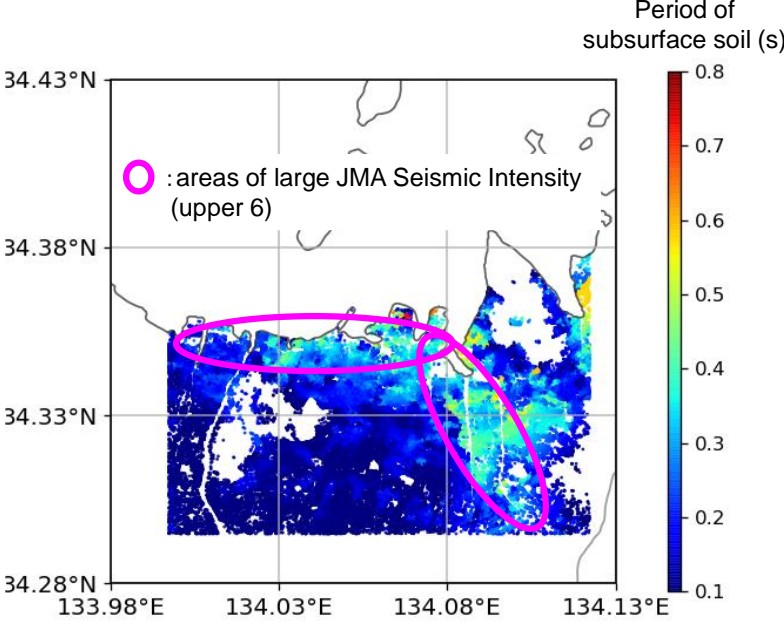

**Figure 3.** Distribution of natural period of the automatically constructed analysis models.

Since the Nankai Trough earthquake is an inter-plate earthquake, it is expected to have large short frequency (or long period) components. A hazard map for this earthquake was released by the local government [20]. It shows places with a greater seismic index, which agrees with the location of longer natural periods of the analysis models. There is a discrepancy between the two, although places with long natural periods are covered by places with greater seismic indices in the hazard map. This means that the overall estimate of earthquake hazard made by the analysis models is in good agreement with the current estimates, but the analysis model used for the physics-based simulations can provide a more reliable estimate for the site of each residential building; at least, we can provide evidence of the reliability of analysis methods that is made from the artificial ground model interpolated by boring data. The seismic index is the seismic intensity scale of the Japan Meteorological Agency (JMA), which is called "shindo" and is strongly correlated with damage of buildings [21].

## 5. Example of Earthquake Disaster Estimation Using Analysis Model of Site Surface Layer

We performed the physics-based simulation of the ground motion amplification and structural seismic response processes for each residential building in the area studied in the preceding section. Analysis models of stratified layers and muti-degree-of-freedom model were automatically constructed for the site surface layer and residential building, respectively.

No errors were reported in the construction of the stratified layer model, suggesting the sufficient robustness of the proposed method. The present physics-based simulation employed the automatic construction method of a multi-degree-of-freedom model for a residential building. This method was verified in a simulation of Tokyo Metropolis Earthquake [13]. The external configuration of the building is used to determine the floor number, area and the type of buildings, wooden houses, reinforced concrete buildings or steel buildings, from which the mass and spring constant of the multi-degree-of-freedom model is constructed to satisfy the empirical relation between the building height and the first natural frequency. The models shown in this paper were constructed by Fujita et al. using the method in [22].

### 5.1. Conditions of Calculation

The physics-based simulation of the ground motion amplification processes uses ground motion at the engineering bedrock as input. We used ground motion provided by the Cabinet Office, Government of Japan, for Nankai Trough Earthquake [23]. The ground motions at the bedrock were given for a square grid of approximately 5 × 4 km, and there were four ground motions in the target area.

Figure 4 presents the four grids of bedrock ground motions together with the acceleration response spectrums calculated by 5% of damping ratio. While ground motion 2 has less components greater than 0.5 s, components between 0.1 and 0.3 are more or less the same for the four ground motions.

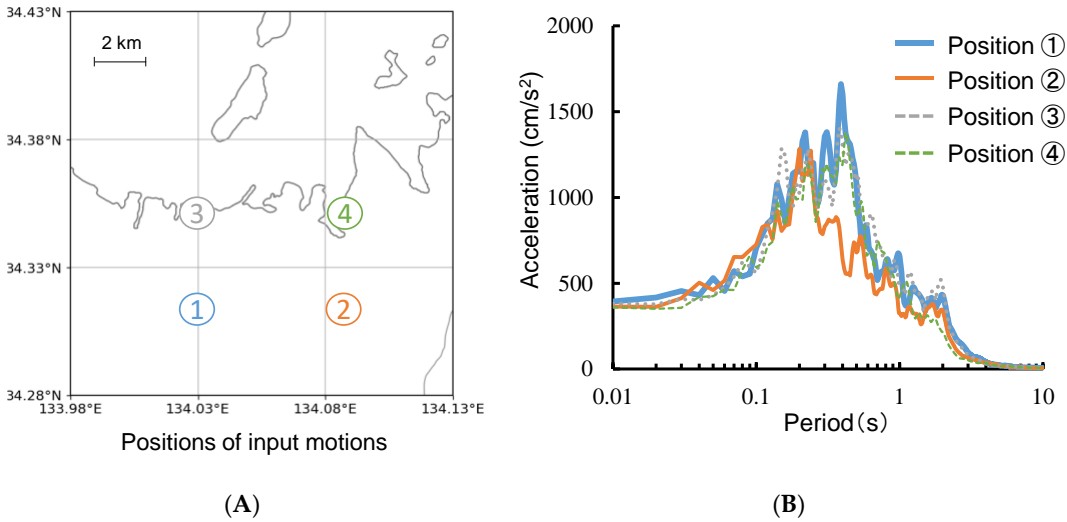

**Figure 4.** Locations of given input motions and acceleration response spectra at each location. (**A**) Locations of input motions at engineering bedrock. (**B**) Acceleration response spectra of input motions.

### 5.2. Details of Analysis Models of Site Surface Layer

Although the simple stratified layer model is employed as an analysis model, we pay attention to the following two points in its detail. The first point is the spatial discretization. A finite element method is used as a numerical analysis program, and the element size of the analysis model is determined to divide wavelength of 10 Hz into 40 elements, which is sufficiently fine in the numerical analysis of linear or nonlinear responses. The wavelength is computed using $V_s$ of each layer.

The second point in the model detail is an extension to the nonlinear analysis using the Ramberg–Osgood model, which is generally used for nonlinear analysis of the ground motion amplification processes. In addition to the shear wave velocity, we need to determine the unit volume weight, reference strain and maximum damping ratio. Unlike the shear wave velocity, these mechanical properties are determined using only the soil type only of the surface layer, as shown in Table 1 [19]. As in the case of the shear wave velocity, it is necessary to classify the soil type into the three categories of sandy, cohesive and gravel soils.

**Table 1.** Parameters of subsurface soil models by the soil classification.

| Soil Classification | Unit Weight (kN/m³) | Reference Strain | Maximum Damping Ratio |
|---|---|---|---|
| Sandy soil | 18.0 | 0.000432 | 0.262 |
| Cohesive soil | 17.5 | 0.00106 | 0.285 |
| Gravel soil | 21.0 | 0.000432 | 0.262 |

### 5.3. Results of Physics-Based Simulations

Seismic response analysis was conducted for all target buildings. Figure 5 depicts visualization of the seismic responses of buildings, whose contour was made by displacement of the buildings. Figure 6 depicts a distribution of the maximum story deformation angle of wooden buildings. Regarding all wooden buildings, the maximum story deformation angle is less than 1/100, which implies that the possibility of large damage on buildings is low.

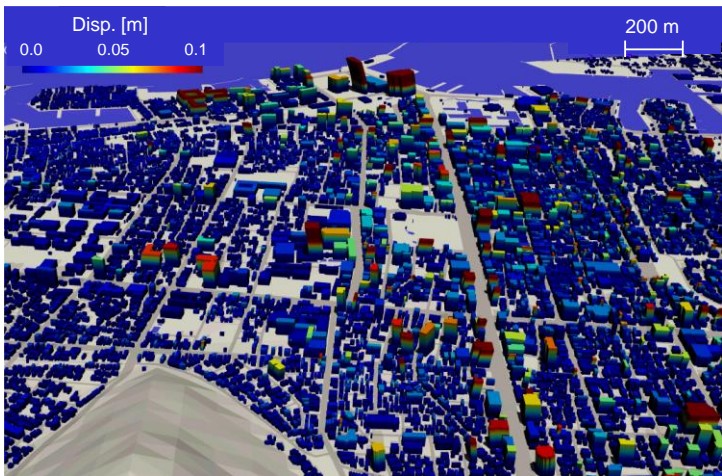

**Figure 5.** Example of visualizing the seismic responses of buildings.

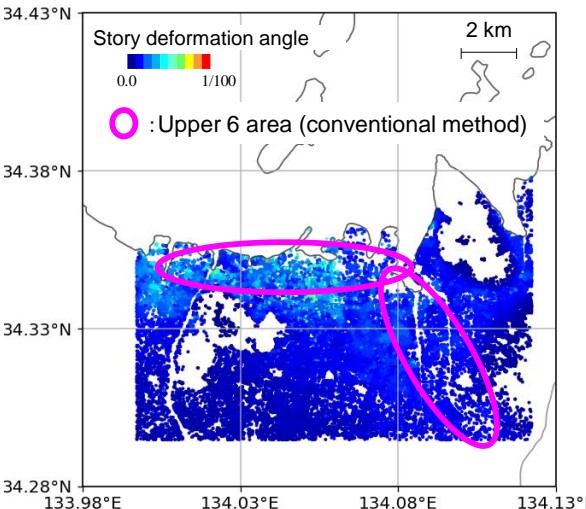

**Figure 6.** Distribution of maximum story deformation angle of wooden buildings.

It was simplified, but the reality and usefulness could be discussed by comparing the results of calculation and by assuming the hazard published by the local government of the area [19]. According to the assumed hazard, the maximum seismic intensity was upper 6, and the area of upper 6 was distributed in limited area, which was consistent with the area of long period of subsurface soil in Figure 3. The responses of buildings of calculation were larger in the area of northwestern part nearby coastline, which were consistent with the upper 6 area of assumed hazard. It was qualitative but could be expected that the results were implied to be realistic. Conversely, the responses of buildings in the area of the eastern part were not consistent with the assumed hazard of upper 6; the calculation results did not show a larger response than the other area. The reason why the calculation results were not larger despite the long period of subsurface soils could be explained by the smaller input motions shown in Figure 4. It was possible that the conventional method used in making the assumed hazard could not achieve such high resolution as the

calculation because it contracted all characteristics of input motions to a single factor of seismic intensity and all characteristics of subsurface soils to a single factor of amplification factor of seismic intensity. Furthermore, the amplification factor was basically estimated by terrain classification of the location, which was given by a much lower resolution than boring data. This difference implies usefulness of the method using calculations, which should be verified more precisely in future works.

With the exception of the difference mentioned above, the local government of the area estimated the number of collapsed buildings to be about 4500, which was different from the calculation results. Although more careful verification is necessary to conclude that the method of this study, which includes the modeling method of subsurface soil, is more useful than the conventional method, the discussion above implies a certain rationality of the method. The method of this study could be more accurate than the simplified method in principle, which could be considered by the above results of high resolution based on the rich data of each location.

## 6. Conclusions

This paper presents the automated construction method of an analysis model site surface layer using a set of boring data of a target area. The analysis model is used in the physics-based simulation of ground motion amplification processes, and the synthesized ground motion can contribute to a more scientifically reliable damage estimation of residential buildings.

The main feature of the proposed method is its robustness. It can be applied to a set of low-quality or a small number of boring data. Reliability of the analysis models is scarified since the method uses only three boring data closest to the target building. However, as in the example problem considered in this paper, the analysis models constructed atomically have no fatal errors. Since a simple layered model is adopted, the analysis model can be used to analyze nonlinear ground motion amplification processes for strong seismic motions in engineering rock masses. In future works, we would like to validate the models constructed by the method and try to clear the accuracy and reliability of the method.

**Author Contributions:** Conceptualization, H.M. and M.H.; methodology, H.M. and M.H.; software, H.M. and M.H.; application, M.H.; writing—original draft preparation, M.H. All authors have read and agreed to the published version of the manuscript.

**Funding:** This research received no external funding.

**Institutional Review Board Statement:** Not applicable.

**Informed Consent Statement:** Not applicable.

**Data Availability Statement:** The data that support the findings of this study are available from the corresponding author upon reasonable request.

**Conflicts of Interest:** The authors declare no conflict of interest.

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
