# Peer review of "Construction and Usefulness Verification of Modeling Method of Subsurface Soil Layers for Numerical Analysis of Urban Area Ground Motion"

_2624-795X, doi:10.3390/geohazards3020013_

Round 1

Reviewer 1 Report

Please check attachment

Reviewer 2 Report

Paper presents a methodology for modeling surface layers by interpolating boring data, and demonstrates its utility, with emphasis on robustness, for urban seismic damage simulations. Overall paper is well organized and informative. I think it can be accepted after minor revisions/editorial edits. Authors should consider the following comments in revising manuscript. 

1) Justification is needed why three boring data is used in the linear interpolation (why not more?). 

2) The type of interpolation the authors use is a moving averaged interpolation, using weighted points in proximity (moving because weights depend on the proximity). Why is this chosen and are there any concerns with smoothness when the number of closest points change? Since only three closest points are used there might be some challenges in getting smooth predictions for f(P) or N(P)?

3) Term position in Figure 4 (and relevant discussion) should be probably replaced with location. 

4) Discussion in Section 5.3 does not clearly demonstrate utility of proposed method (discussion related to Figures 5 and 6 is somewhat simplistic for example). Can authors enhance it? 

Author Response

This manuscript is a resubmission of an earlier submission. The following is a list of the peer review reports and author responses from that submission.

Round 1

Reviewer 1 Report

Motoyama and Hori have proposed the manuscript "Construction and Usefulness Verification of Modeling Method of Subsurface Soil Layers for Numerical Analysis of Urban Area Ground Motion" for publication in GeoHazards.

Unfortunately, the proposed manuscript is unsuitable for publication and must be rejected.

Let's start with two general considerations.

1) I am not a native English speaker, but I am sure that the English used in the manuscript is deficient and this often makes reading complicated.

2) The manuscript is a technical note and not a scientific article. This is also demonstrated by the bibliography, which is deficient and limited to local publications. The authors seem to ignore what is published in the rest of the world.

Let's move on to the strong criticalities related to the scientific aspects.

The authors' considerations on modeling buried geological bodies are inconsistent and disconnected with scientific literature. Decades of technical and scientific advances in the field of numerical modeling, particularly for oil and gas exploration and production, are completely ignored by the authors. Not only that, even the most elementary principles of stratigraphy, now hundreds of years old, are ignored. The authors talk about consistency and robustness of the model but then ignore the inconsistency of an ultra-simplified approach, according to which well logs can be correlated in groups of three, using linear interpolators and IDW! How can a large area risk model be consistent if the subsurface model is not spatialized: there is no modeling parameter that the authors use to spatialize, each M=3 model is consistent only with itself and disjointed from adjacent models. Spatial correlation algorithms, such as precisely Kriging (and many others), exist for this very reason. Much better then to disregard data location at all and use other methods of estimating dynamic parameters to assess seismic amplification. In this regard, just in Japan the authors can refer to the work of Iwahashi et al. on the estimation of Vs30.

And we come to the aspects related to seismic risk estimation.

First, throughout the work the concepts of hazard and risk are used incorrectly, often being mistaken for each other. The modeling of the hazard before and the risk after are strongly lacking and it is obscure how the authors consider evaluating the reliability of their risk estimate.

In conclusion, I suggest that the authors reconsider publishing this work. I do not believe the manuscript can be improved because of the overall deficiencies of the work.

Reviewer 2 Report

The paper is quite interesting, especially for potential contribution of the proposed method to the construction, in quick and automatic way by using borehole data, of the subosoil stratigraphic model for studies about seismic risk assesment for urban planning and/or building design, even if were not taken into account bidimensional conditions.

However some sections (3 ,4 and 5) could be further detailed to valorize and explain the proposed method and to make better comprehension of the results from readers. In particular you can consider the following modifies and suggestions:

line 165-166: “Moreover, rationality of the constructed model is considered by using the period estimated by the model”

Specify better how this rationality was confirmed or, for example, it could be useful to present, here or later, some stratigraphic columns from the study area and their obtained fundamental period.

Line 172-176: insert, for clearness and to show the robustness of the method, some examples of subsurface soil model obtained for target positions by the stratigraphic boreholes data used.

Line 181: were only two soil units defined/used for all study area? Specify N what represents and the bibliographic references for these equations

Line 184: it would be useful to insert an extract from the seismic hazard map cited

Line 185: “earthquake intensity”: do you mean macroseismic intensity? If yes, according what scale of measurement? MSC, SHINDO, EMS, MM etc.

Line 182-186: to verify the correctness of the constructed model you should compare for some check point, the stratigraphy of a real borehole and that one derived from the model. To find in some areas a period/frequency fundamental of vibration means only to have conditions of amplification for site effects.

The only presence of low fundamental periods or no resonance, is not sufficient to giustify the rationality of the geological model of the subsoil constructed, besides, it is not clear the link of these fundamental periods with risk of large eartquake in the area (are maybe present in that area seismogenic faults to generate strong earthquakes?); this part must be improved to make it clearer.

Line 190-200: in this part it would be also useful to show for some points (for example 1,2,3 and 4 of Fig.6) the results of the seismic response of the soil by comparing output spectra (on the ground level) vs input spectra (on the bedrock)

Line 229: specify by the acronym the scale used for earthquake intensity

Line 236-237: Could it be relevant the different type of buildings in terms of structural characteristics, period of vibration (for example for different height respect those ones present in the northwestern part) etc.?

Fig. 4: insert in the figure: geographic Nord and a box which shows the position of this area respect to all Japan

Fig. 5: enalarge size of the figure (for better readability) and insert: graphic scale and geographic Nord; specify acronym of the Intensity scale

Fig. 6A: insert: graphic scale and geographic Nord

Fig. 6B: preferably use g or cm/s2 for accelaration unit and in the caption: spectra instead of spectrums

Fig. 7: insert: graphic scale and geographic Nord; enlarge the size of displacement legend (especially characters are not so readbly)

Fig. 8: enalarge size of the figure (for better readability) and insert: graphic scale and geographic Nord

Tab.1: insert if possible also the range of Vs for the different soils